# NeuroSAFE robot-assisted laparoscopic prostatectomy versus standard robot-assisted laparoscopic prostatectomy for men with localised prostate cancer (NeuroSAFE PROOF): protocol for a randomised controlled feasibility study

Eoin Dinneen,[1,2] Aiman Haider,[3] Clare Allen,[4] Alex Freeman,[3] Tim Briggs,[1] Senthil Nathan,[1] Chris Brew-Graves,[2] Jack Grierson,[2] Norman R Williams,[2] Raj Persad,[5] Neil Oakley,[6] Jim M Adshead,[7] Hartwig Huland,[8] Alexander Haese,[8] Greg Shaw[1,2]

For numbered affiliations see end of article.

**Correspondence to**
Eoin Dinneen;
eoindinneen@gmail.com

## ABSTRACT

**Introduction** Robot-assisted laparoscopic prostatectomy (RALP) offers potential cure for localised prostate cancer but is associated with considerable toxicity. Potency and urinary continence are improved when the neurovascular bundles (NVBs) are spared during a nerve spare (NS) RALP. There is reluctance, however, to perform NS RALP when there are concerns that the cancer extends beyond the capsule of the prostate into the NVB, as NS RALP in this instance increases the risk of a positive surgical margin (PSM). The NeuroSAFE technique involves intraoperative fresh-frozen section analysis of the posterolateral aspect of the prostate margin to assess whether cancer extends beyond the capsule. There is evidence from large observational studies that functional outcomes can be improved and PSM rates reduced when the NeuroSAFE technique is used during RALP. To date, however, there has been no randomised controlled trial (RCT) to substantiate this finding. The NeuroSAFE PROOF feasibility study is designed to assess whether it is feasible to randomise men to NeuroSAFE RALP versus a control arm of 'standard of practice' RALP.

**Methods** NeuroSAFE PROOF feasibility study will be a multicentre, single-blinded RCT with patients randomised 1:1 to either NeuroSAFE RALP (intervention) or standard RALP (control). Treatment allocation will occur after trial entry and consent. The primary outcome will be assessed as the successful accrual of 50 men at three sites over 15 months. Secondary outcomes will be used to aid subsequent power calculations for the definitive full-scale RCT and will include rates of NS; PSM; biochemical recurrence; adjuvant treatments; and patient-reported functional outcomes on potency, continence and quality of life.

**Ethics and dissemination** NeuroSAFE PROOF has ethical approval (Regional Ethics Committee reference 17/LO/1978). NeuroSAFE PROOF is supported by National Institute for Healthcare Research Research for Patient Benefit funding (NIHR reference PB-PG-1216-20013).

### Strengths and limitations of this study

► This is the first feasibility clinical trial to compare NeuroSAFE robot-assisted laparoscopic prostatectomy (RALP) to a UK 'standard of care' RALP.
► Multicentre, randomised controlled trial design.
► This is the protocol for a feasibility study, and therefore this study is not powered to allow for the analysis of secondary outcomes.
► Secondary outcomes include validated patient-reported outcome questionnaires, histological and oncological end points, and health economics.

Findings will be made available through peer-reviewed publications.
**Trial registration number** NCT03317990.

## INTRODUCTION

Nerve sparing (NS) robot-assisted laparoscopic radical prostatectomy (RALP) is associated with superior postoperative functional outcomes such as erectile function and possibly urinary continence.[1 2] While functional results after RP are of importance to many men, the primary objective of a cancer operation remains complete eradication of the tumour.[3] Therefore, it is important that performing NS RALP does not compromise that oncological outcome. Positive surgical margins (PSMs) are associated with greater risk of biochemical recurrence,[4] adjuvant therapies (which negate any improved functional outcomes following NS RALP) and disease progression. As such, despite the improved anatomical understanding and

technological advancement of the robotic platform, NS RALP is often eschewed in favour of assuring the safety of a negative surgical margin by performing wide excision around the prostate. Uncertainty in this area is compounded by the fact that the accuracy of preoperative imaging techniques and physical examination to detect extracapsular extension (ECE) and/or neurovascular cancer involvement are debatable.[5] In particular, pooled data from a recent diagnostic meta-analysis found MRI to have a limited sensitivity of 0.57 (95% CI 0.49 to 0.64) when predicting ECE.[6] Therefore, RALP can often lead to unwarranted sacrificing of important functioning neurovascular bundles (NVBs). When deciding whether to perform NS RALP or non-NS RALP, surgeons rely on parameters such as preoperative erectile function, D'Amico Risk Classification, radiological staging, and location and volume of tumour to cautiously assess the safety of an NS approach. These assessments may not give a true picture and are prone to subjective evaluation. The concept of a frozen section-navigated NS during RALP using neurovascular structure adjacent frozen section examination of the prostate resection margin (NeuroSAFE) has been described by the Martini-Clinik in Hamburg, Germany.[5 7 8] These authors and others report benefit in functional outcomes and improved oncological safety in their series[9 10] though other retrospective series are not as clear-cut.[11]

The NeuroSAFE technique has not yet been widely adopted, as concerns remain that it is time-consuming and resource-consuming, has low sensitivity and specificity and has potentially conflicting oncological results.[12–15] Neither intraoperative fresh-frozen section (FFS) in RALP nor the NeuroSAFE technique have been prospectively evaluated by a randomised controlled trial (RCT). Moreover, few studies have assessed the impact of FFS during RALP on longer term patient outcomes such as biochemical recurrence, adjuvant cancer treatments (such as radiotherapy and hormones) and comprehensive functional outcomes.

## RESEARCH NEED
To determine whether the NeuroSAFE technique (FFS of the prostate tissue adjacent to the NVBs) during RALP is helpful to surgical teams (and therefore patients) who are balancing the competing goals of cancer control and functional optimisation. An attempt to answer this question will require a multidimensional approach focusing on preoperative and operative parameters, final histological outcomes, adjuvant treatments, quality of life, erectile function, urinary continence and health economics. There is recognition that surgical RCTs can be hard to recruit to and that patients may not accept their allocated treatment option.[16] For this reason, we propose to undertake a feasibility study to examine recruitment rates, acceptance of allocated treatment and to rehearse collection of outcomes.

## STUDY AIMS AND OUTCOMES
The aim is to prospectively recruit for randomisation eligible patients to either standard RALP (control arm) or NeuroSAFE RALP (intervention arm). This feasibility trial has a single-blinded, 1:1 randomised design. This article reports the protocol (v.2.0, 6 February 2018) for the NeuroSAFE PROOF trial and follows SPIRIT (Standard Protocol Items: Recommendations for Interventional Trials) reporting guidelines.[17]

The trial objectives are to assess the feasibility and acceptability of:
► Recruiting men with localised prostate cancer to an RCT of NeuroSAFE RALP versus standard RALP.
► Collecting data for outcome measures, including patient-reported outcomes.
► Estimating treatment effects to inform power calculations for the definitive full-scale future trial.
► The study's procedures, interventions and follow-up regimen among men being treated with RALP for localised prostate cancer.

The following criteria will have to be met to proceed to a full-scale trial:
► Recruitment of 50 men over 15 months from opening. Fifty men was decided on to demonstrate that if similar recruitment rates were maintained in the full-scale NeuroSAFE PROOF study, the trial would be able to recruit the several hundreds of men likely necessary to appropriately power the said trial over the course of approximately 2–3 years.
► Recruitment and performance of procedures (both intervention and control) as per allocation at three prespecified participating sites (UCLH, Bristol and Sheffield). At least two treatments (one intervention, one control) should be performed at each site.
► Methodological or practical issues with the trial design should be identified and amended before full-scale trial.
► Good acceptability of the intervention among patients and their families, indicated in qualitative feedback and public and patient involvement (PPI) events.
► Acquisition of comprehensive patient-reported outcomes measure including health economics questionnaires.

## PUBLIC AND PATIENT INVOLVEMENT
Patient feedback on the design of the study was obtained at two NeuroSAFE PROOF PPI sessions on 12 July 2018 and 20 September 2018. The second event was attended by men participating in NeuroSAFE PROOF. The PPI events were supported by Macmillan Cancer (Charity no 261017) and Orchid (Charity no 1080540). Participants, patients and their families were asked specifically about the level of blinding, the burden of follow-up appointments and priorities in their recovery from RALP. Following their feedback, NeuroSAFE PROOF now informs men following surgery of their NS status, though blinding to allocation status (intervention or control) is maintained. Furthermore, men expressed

keen preference to know their treatment allocation once exiting the 12 months of follow-up period, and this is now incorporated into trial design. Patient representatives sit on the trial steering committee (TSC) for NeuroSAFE PROOF and share oversight of the management of the trial. The study is also funded by National Institute for Healthcare Research Research for Patient Benefit (NIHR RfPB) stream, which has patient members on their decision panels. On completion of NeuroSAFE PROOF, prostate cancer patient groups will be consulted again on amendments to the design of the full-scale RCT. The results will be published following peer review, and anonymised data will be presented at national and international conferences.

## METHODS AND ANALYSIS
### Trial design
NeuroSAFE PROOF feasibility study is a prospective, multicentre, feasibility RCT in patients undergoing RALP for localised prostate cancer. Eligible patients will be consented and randomised 1:1 to NeuroSAFE RALP (intervention) or standard RALP (control) after multidisciplinary team (MDT) review in National Health Service (NHS) urological cancer centres. It is not possible to blind the surgical team to the treatment received on the day of surgery. Researchers for whom knowledge of allocation is imperative, that is, those involved in operating on patients or coordinating operating lists or pathology teams are not blinded to treatment allocation, other members of the research team are blinded to treatment allocation. Participants are not informed of treatment allocation until completing 12 months of follow-up and exiting the study, though they are informed of their ultimate nerve spare (NS) status (ie, no NS, unilateral NS, bilateral NS). The primary outcome is feasibility of recruitment.

Secondary outcomes will include:
► Rates of NS performed during RALP.
► Rates of PSMs.
► Adjuvant therapies and biochemical recurrence.
► Patient-reported outcome questionnaires assessing potency, urinary continence and quality of life.
► Patient-reported healthcare resource diaries.

These outcome measures will allow us to explore the feasibility and acceptability of delivering a full-scale multicentre RCT. The decision to include 50 feasibility study patients in the full-scale NeuroSAFE PROOF trial will only be allowed if the feasibility study aligns sufficiently closely and will be at the discretion of the independent TSC.

### Trial population
Prior to entry, patients must be accurately staged (eg, multiparametric MRI (mpMRI) prostate and cross-sectional imaging to assess for distant metastases (eg, bone scan or whole-body MRI)), within 3 months prior to randomisation. Eligible patients must have had their case discussed at NHS cancer MDT and deemed suitable and fit for RALP. Eligible participants will fulfil all the inclusion criteria and none of the exclusion criteria as defined below:

### Inclusion criteria
1. Men opting to undergo RALP for organ-confined prostate cancer.
2. Potent men (International Index of Erectile Function (IIEF) 22–25 not using phosphodiesterase type 5 (PDE-5) inhibitors or other medications or vacuum pump).
3. Men who are continent of urine (no self-reported urinary incontinence).
4. Able to give written informed consent to participate.

### Exclusion criteria
1. Unable to undergo RALP.
2. Known overactive bladder.
3. Previous treatment for prostate cancer.
4. Previous/current hormone treatment for prostate cancer.
5. NS deemed futile due to locally advanced disease by surgeon and radiologist.

An overview of the study schema can be seen in figure 1.

### Sample size
The primary outcome of NeuroSAFE PROOF is to demonstrate adequate recruitment to prove feasibility of the full-scale definitive NeuroSAFE PROOF RCT. Operative data, preliminary functional outcomes data and preliminary oncological outcomes data from this feasibility data will be used to help determine power calculations for the full-scale NeuroSAFE PROOF RCT. Previous literature suggests that 80% of men undergoing bilateral NS will have erections sufficient for penetrative sex, 40% of men undergoing unilateral NS and 10% of men undergoing no NS.[18]

### Recruitment
NeuroSAFE PROOF will recruit patients attending NHS cancer centres. All patients who have a diagnosis of prostate cancer and who have been recommended for RALP by a specialist NHS regional MDT meeting will be eligible to be approached.

### Consent
Written informed consent will be obtained from each patient prior to study entry and performing baseline trial assessments. An ethics committee-approved patient information sheet will be provided to facilitate this process. Prospective participants will be given at least a week to read the patient information sheet prior to being reapproached with regards to recruitment. The investigator, or their designee, must ensure adequate explanations of the trial that participation is voluntary and they can withdraw at any time. In consenting to the trial, participants understand that they are consenting to provide study follow-up and data collection. A patient may withdraw from the trial at any time without prejudice to his subsequent treatment.

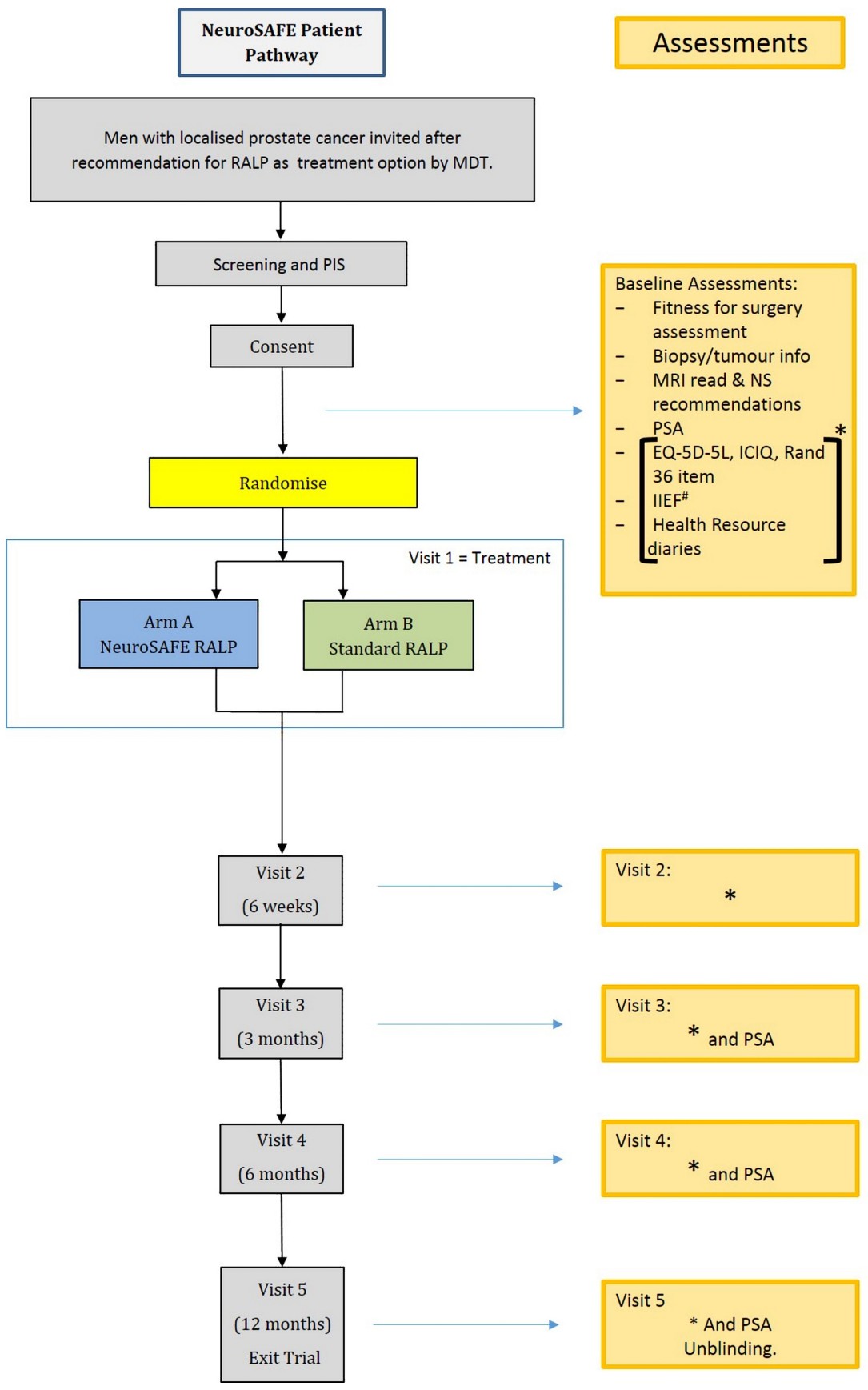

**Figure 1** NeuroSAFE PROOF feasibility study schema. EQ-5D-5L, EuroQuol-5 Dimension-5 Level Questionnaire; ICIQ, International Consultation on Incontinence Questionnaire; IIEF, International Index of Erectile Function; MDT, multidisciplinary team; NS, nerve sparing; PIS, Participant Information Sheet; PSA, Prostate Specific Antigen; RALP, r obot-assisted laparoscopic prostatectomy.

## Randomisation

Patients will be randomised using an online system (https://www.sealedenvelope.com/trials/) on a 1:1 basis to either NeuroSAFE RALP or standard RALP. A computer-generated adaptive minimisation algorithm that incorporates a random element will be used to ensure treatment groups are balanced (stratified) for centre. Treatment allocation will occur after trial entry and consent. Participants will not be informed of their treatment allocation until exiting the trial 12 months following their surgery. The clinical teams performing and coordinating surgery will not be blinded to treatment allocation as this is impractical, and any members of the research team not involved in these activities will be blinded.

## SETTING

Participants will be recruited from NHS cancer centres undertaking RALP who have the ability to perform the additional NeuroSAFE technique. Recruiting sites will be invited by the trial management group (TMG). Trial sites will have well-developed RALP programmes, routinely performing at least 250 cases per year and undergoing satisfactory NHS quality assurance and safety visits.

## Surgeon and unit accreditation

Variations in surgical team performance can produce differences in outcomes from RALP.[19] As such, to minimise this potential source of confounding, surgeons and surgical teams participating in NeuroSAFE PROOF feasibility study will require accreditation from the TMG. Further, surgeons performing trial treatment need to have completed more than 100 cases and have submitted these data to the BAUS Oncology database.

## Robot-assisted laparoscopic prostatectomy

Patients will undergo RALP using the DaVinci surgical system as is standard of care in the NHS. All patients will undergo a preoperative mpMRI that will be interpreted by a consultant genitourinary radiologist with at least 2 years experience in reading prostate mpMRIs. The preoperative mpMRI will be interpreted by the radiologist with biopsy information and will be used to evaluate presence of cancer and likelihood of ECE in zones according to the PIRADS anatomic division of the prostate at the base, the mid-gland and the apex. In each zone, using a 1–5 scale (1, definitely absent; 2, probably absent; 3, possibly present; 4, probably present; 5 definitely present), they will record the likelihood of tumour on each side. Using the same 1–5 scale they then indicated the likelihood of ECE in each corresponding zone as has been previously done by Akin et al.[20] Subsequently, the radiologist using the mpMRI makes an NS recommendation for each side of the prostate for each participant regardless of treatment arm allocation. The radiological NS recommendation will be recorded:
► NS: Yes.
► NS: No.
► Digital rectal examination dependent.

## Control arm: standard RALP

Standard RALP (control arm) is performed as per NHS routine practice. Preoperative parameters used to guide surgeon NS decision include mpMRI review with genitourinary radiologist recommendation with regards NS, prostate biopsy histology and digital rectal examination under general anaesthesia. Individual surgeons are asked after RALP to grade the quality of NS performed on each side numerically as seen below as previously described[21]:
► Grade 4: No NS. Wide excision of lateral pelvic fascia (LPF) and Denonvilliers' fascia.
► Grade 3: Limited NS or partial/incremental NS. Incision through outer compartment of LPF.
► Grade 2: Interfascial NS. LPF is taken just outside the layer of the veins of the prostate capsule. Still largely preserving the large neural trunks (also known as the NVBs).
► Grade 1: Intrafascial NS. LPF is taken just outside the prostate capsule. Represents greatest possible NS.

Detailed times of starting the RALP and finishing the RALP are recorded on the day of surgery to calculate the length of each case.

## Intervention arm: NeuroSAFE RALP

NeuroSAFE RALP (intervention arm) will be performed in accordance with previously described methods, initially developed at the Martini Clinic, Hamburg, Germany.[5 8 22] The additional steps outlined include NS technique and apical dissection, specimen removal, intraoperative frozen section protocol, simultaneous urethravesical anastomosis (with/without pelvic lymphadenectomy where performed), pathological processing of specimen, pathology-reporting protocol and secondary excision of the NVB (where appropriate). Detailed results of the FFS examination will be collected and included in the results, including number of sections positive, length of positive margin, identity and grade of pathologist. When the frozen section examination demonstrates cancer at the margin of the prostate as per pathology-reporting protocol, secondary excision of the NVB is described by the surgeon in one of three ways: (1) No tissue resected, (2) Local excision of Denonvilliers'/periprostatic fascia or (3) Entire bundle resected. Secondarily resected tissue (after FFS pathology phone call, when performed) is sent for routine paraffin-embedded histological analysis and is not analysed as part of the intraoperative FFS. Detailed times of the beginning of the RALP, the removal of the prostate for specimen painting, arrival of specimen in laboratory, communication of details of FFS to the surgical team and finishing the RALP are recorded on the day of surgery.

Participating sites all visited the central site (UCLH) prior to their Site Initiation Visits to receive teaching and standardisation in the surgical and histopathological aspects of NeuroSAFE RALP (intervention arm). Subsequently, researchers from the central site (GS and

**Table 1** Table of assessments

| | Baseline/ recruitment | Visit 1 treatment | Visit 2 (6 weeks post-RALP) | Visit 3 (3 months) | Visit 4 (6 months) | Visit 5 (12 months) |
|---|---|---|---|---|---|---|
| Informed consent | x | | | | | |
| Randomisation | x | | | | | |
| PSA | x | | | x | x | x |
| Standard RALP or NeuroSAFE RALP | | x | | | | |
| Adverse events | | x | x | | | |
| EQ-5D-5L, ICIQ, Rand 36 | x | | x | x | x | x |
| IIEF | x | | x | x | x | x |
| Adjuvant therapies | | | x | x | x | x |
| Health resource diary | | | x | x | x | x |

ICIQ, International Consultation on Incontinence Questionnaire; IIEF, International Index of Erectile Function; RALP, robotassisted laparoscopic prostatectomy. ICIQ, International Consultation on Incontinence Questionnaire; IIEF, International Index of Erectile Function; RALP, robot-assisted laparoscopic prostatectomy.

AH) reciprocated the visit for the first NeuroSAFE RALP performed by each site to ensure fidelity to technique protocol.

## Data collection

Post-treatment trial assessments will be conducted at follow-up appointments. All patients will have follow-up appointments at 6 weeks following surgery, 3 months, 6 months and finally 12 months following their treatment. Table of assessments is demonstrated below (table 1).

## Time points

1. Baseline/preoperative: at the time of consent, trial entry and randomisation to treatment allocation.
2. Visit 1: operative parameters recorded and any immediate postoperative complications/adverse events.
3. Outpatient follow-up: visits 2, 3, 4 and 5 will record patient-reported outcome measures and healthcare resource diaries. Adjuvant treatments and oncological outcomes will be recorded prospectively alongside functional assessments.
4. On visits 3, 4 and 5, a serum PSA will be taken in addition to functional questionnaires and adjuvant treatment outcomes.

## Secondary end point measures

Secondary end point measures include:

1. IIEF-15 (baseline, 6 weeks, 3 months, 6 months and 12 months): a self-completion tool for men focusing on erectile function and sex life. Measured domains include erectile function, orgasmic function, sexual desire, intercourse satisfaction and overall satisfaction.[23]
2. Rand 36-Item Health Survey (baseline, 6 weeks, 3 months, 6 months and 12 months): a self-completion questionnaire that laps eight concepts: physical functioning, bodily pain, role limitations due to health problems, role limitations due to personal or emotional problems, emotional well-being, social functioning, energy/fatigue and general health perceptions.[24]
3. International Consultation on Incontinence Questionnaire (ICIQ) (baseline, 6 weeks, 3 months, 6 months and 12 months): a self-completion tool for patients to subjectively measure frequency and severity of urinary loss, and impact on quality of life for those with urinary incontinence.[25]
4. EQ-5D-5L (baseline, 6 weeks, 3 months, 6 months and 12 months): a self-completion tool for patients that is applicable to a wide range of health conditions and treatments. Measured domains include mobility, self-care, usual activities, pain and anxiety or depression.[26]
5. Health resource diaries (6 weeks, 3 months, 6 months, 6-month visit diary will be returned at the 12-month visit). This will allow the collection of resource use data from point of operation until trial exit at 12 months. These diaries are non-validated.
6. Postoperative: adverse events and complications will be recorded. Clavien–Dindo classification of surgical complications will be used to assess for any surgical complications as per normal hospital practice.
7. Histology: following RALP, the following details will be recorded as per standard histological analysis of prostatectomy mount: histological type, Gleason grade, Gleason group, tumour volume, extraprostatic extension, seminal vesicle involvement, lymphovascular invasion, description of margin involvement (including apical, basal, circumferential), tumour stage, nodes, PSMs.
8. Oncological outcomes (3 months, 6 months and 12 months): the curative outcomes from RALP will be examined to determine local and distant recurrence, metastases, PSA and biochemical recurrence, need for adjuvant therapies and survival (overall and cancer specific).

## STATISTICAL ANALYSIS

As NeuroSAFE PROOF is a feasibility trial, there is no intention to undertake detailed statistical analysis. Preliminary analysis will be performed after five cases have reached 'visit 3' to rehearse data extraction, completeness of follow-up, fidelity of data and by proxy acceptability of follow-up measures. Further preliminary data analysis, maintaining blinding, of the secondary outcomes 'margin status' and 'RALP NS status' will be performed by the data monitoring committee (DMC) after 40 surgeries have been performed to evaluate and help revise power estimations for the full-scale RCT. Potential bias due to missing data will be investigated by comparing descriptively the baseline characteristics of the trial participants with complete outcome measurements to those who have missing outcome measurements. Men will be offered the option of telephone follow-up and/or be sent questionnaires by post if they are unable to attend clinic appointments for follow-up. Additionally, patients wishing to withdraw from the trial will be counselled regarding end of active participation, as this will allow the trial team to continue to use their outcome data for an intention-to-treat analysis. Records will be kept of all participants allocated to a treatment arm who do not undergo allocated treatment with explanatory notes. These instances will be highlighted to the Surgical & Interventional Trial Unit at University College London (study sponsor) (SITU) and the TSC for judgement on whether inclusion in outcomes is appropriate.

## SAFETY

The number of adverse events related to serious adverse events (SAEs) will be summarised descriptively by arm, by grade and body system. RALP is a major surgery that has a number of recognised complications and a very low risk of death (less than 1 in 100). Operative/postoperative RALP complications will be graded using the Clavien–Dindo classification. The central trial management team will ask sites to submit complication data blinded by arm of treatment. This will be assigned Clavien–Dindo classification centrally.[27] All SAEs will be recorded in the medical records, the case report form, the sponsor's adverse event log and an SAE form. The site principal investigator (PI) or designated individual will complete an SAE form, and the form will be sent to SITU within 5 working days of becoming aware of the event. The study chief investigator or site PI will respond to any SAE queries raised by the sponsor as soon as possible. Where the event is unexpected and thought to be related to the procedure, this must be reported by the PI to SITU, who will then inform the Health Research Authority within 15 days.

## DATA MONITORING

This trial will use an electronic case report form (eCRF), and trial data will be entered into an approved, protected database (https://neurosafe.slms.ucl.ac.uk). Access to the eCRF system will only be provided to staff with the appropriate authority. Participants will be given a unique number and subject identifier. Data will be entered under this identification number onto the central database stored on the servers. The database will be password protected and only accessible to members of the NeuroSAFE study team as well as external regulators if requested. The servers are protected by firewalls and are patched and maintained according to best practice. The physical location of the servers is protected by CCTV and security door access. The database software provides a number of features to help maintain data quality, including: maintaining an audit trail, allowing custom validations on all data, allowing users to raise data query requests and search facilities to identify validation failure/missing data. After completion of the study, the database will be retained on the servers of University College London for ongoing analysis of secondary outcomes. The identification, screening and enrolment logs, linking participant identifiable data to the pseudoanonymised subject numbers will be held in written form in a locked filing cabinet. After completion of the study, sites will store screening and enrolment logs securely for 10 years.

## TRIAL FUNDING, ORGANISATION AND ADMINISTRATION

The trial was developed by the NeuroSAFE PROOF TMG and has been funded by University College London Hospitals NHS Foundation Trust (UCLH), The Rosetrees Foundation and the NIHR Research for Patient Benefit (RfPB) stream (reference: PB-PG-1216-200113). Applied Medical are contributing disposable laparoscopic trocar ports suitable for use in NeuroSAFE RALP (intervention arm), but the company has had no role in trial design and will have no role in trial implementation, analysis, interpretation or writing any reports. The trial is sponsored by University College London and has registered sponsor reference number 17/0443 and ClinicalTrials.gov (NCT03317990) on 23 October 2017 with an amendment made on 1 June 2018. All members of the trial are Good Clinical Practice trained. A DMC will monitor patient safety and the rate of recruitment of subjects in the study. They will meet at least once a year while the trial is ongoing for routine review of safety data and trial progression. They have power to call additional meetings and review data at any point in the trial should they wish to do so. The DMC may report their findings to the TSC. The TSC is an independent committee consisting of relevant, experienced clinicians and researchers. The TSC will ensure the study is conforming to governance requirements as set out by the trial sponsor. The TSC will meet at least once a year. The sponsor may also arrange an independent trial monitor to review the study data.

## ETHICS AND DISSEMINATION

Ethical approval for NeuroSAFE PROOF was granted on 6 February 2018 (regional ethics committee (REC)

reference 17/LO/1978). Here, we report version 2.0 of the protocol. The sponsors, health resources approval (HRA) body and REC will approve any future amendments as appropriate. Similarly, all participating sites have (or will have) gained local REC prior to receiving a site initiation visit and being given the permission to open recruitment.

Non-blinded results of the study will be published in peer-reviewed publications and will be presented at relevant national and international conferences. The TMG will not present the arms in comparison to one another to avoid loss of equipoise and introduction of bias into the full-scale RCT. The TMG will work with a patient panel to develop lay reports to disseminate research findings to patient groups and the clinical teams at participating sites.

## DISCUSSION

Intraoperative FFS analysis of the NVB adjacent prostate margin during RALP to guide NS is now an established technique in a number of centres. Published large series from these centres demonstrate improvements on their outcomes, both functional and oncological. In spite of the possible benefit to men with localised prostate cancer undergoing surgery, the NeuroSAFE technique during RALP has not been widely introduced in the UK. The lack of level 1 evidence to support NeuroSAFE RALP is a valid reason for this.

The NeuroSAFE PROOF RCT feasibility study will be the first trial to assess the feasibility of conducting a randomised trial to evaluate intraoperative frozen section evaluation of the prostate margin during RALP anywhere in the world. The results of this feasibility trial will be used to prepare the full-scale NeuroSAFE PROOF RCT.

### Trial status

NeuroSAFE PROOF RCT feasibility study opened to recruitment in April 2018 using protocol version 2.0 (6 February 2018) and is due to close to recruitment in January 2020 or after the 50th patient is consented and randomised. NeuroSAFE PROOF RCT feasibility study will therefore close in January 2021 or when the last participant to undergo treatment completes the 12-month follow-up as per protocol. Amendments were reviewed and approved by the sponsor and the REC. Protocol amendments are disseminated to relevant parties by SITU.

### Author affiliations
[1]Department of Urology, University College Hospital London, London, UK
[2]Division of Surgery & Interventional Science, University College London Medical School, London, UK
[3]Department of Histopathology, University College Hospital London, London, UK
[4]Department of Radiology, University College Hospital London, London, UK
[5]North Bristol NHS Trust, Westbury on Trym, Bristol, UK
[6]Sheffield Teaching Hospitals NHS Foundation Trust, Sheffield, Sheffield, UK
[7]Department of Urology, Lister Hospital, Stevenage, UK
[8]Martini Klinik, Department of Urology, University Hospital Eppendorf, Hamburg, Germany

**Acknowledgements** We gratefully thank the participants, PIs, research nurses, all the clinicians involved in providing care, regional cancer service coordinators, data managers and other site staff who have been responsible for setting up, recruiting participants and collecting the data for the trial. We are grateful for the trial oversight provided by the sponsor and the members of the TSC. The TSC members are Jack Cuzick (Chair), Queen Mary's University of London, Alastair Lamb, Oxford, Imran Ahmad, Glasgow and Abay Mulatu, Coventry.

**Contributors** Conception and design of NeuroSAFE PROOF trial: GS, ED, AH, JG, AF, TB, SN, CA, JMA, HH, AH, NRW, RP, NO and CB-G. Writing of the manuscript: ED, AH, CA, AF, RP, NO, CB-G, JG and GS. All authors have read and approved the final manuscript. The trial will comply with the authorship criteria recommended by the International Committee of Medical Journal Editors.

**Funding** The trial was funded by the NIHR RfPB and The Rosetrees Foundation. ED is funded by the NIHR RfPB.

**Competing interests** Within NeuroSAFE PROOF, laparoscopic ports are supplied by Applied Medical but Applied Medical has had no role in the design, analysis or collection of the data; in writing of the manuscript; or in the decision to submit the manuscript for publication.

**Patient consent for publication** Not required.

**Provenance and peer review** Not commissioned; externally peer reviewed.

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
