## [Reviewer comments · BMJ Open]

ARTICLE DETAILS

TITLE (PROVISIONAL)	NeuroSAFE Robot Assisted Laparoscopic Prostatectomy versus standard Robot Assisted Laparoscopic Prostatectomy for men with localized prostate cancer (NeuroSAFE PROOF): protocol for a randomised controlled trial feasibility study.
AUTHORS	Dinneen, Eoin; Haider, Aiman; Allen, Clare; Freeman, Alex; Briggs, Tim; Nathan, Senthil; Brew-Graves, Chris; Grierson, Jack; Williams, Norman; Persad, Raj; Oakley, Neil; Adshead, Jim; Huland, Hartwig; Haese, Alexander; Shaw, Greg

VERSION 1 - REVIEW

REVIEWER	Eva Haglind Department of Surgery, Sahlgrenska University Hospital, Göteborg, Sweden
REVIEW RETURNED	06-Dec-2018

GENERAL COMMENTS	I think the title contains a language problem - feasibility study RCT – where “study” should be deleted, as trial is hidden behind RCT. A good example of “danger” with abbreviations. Abbreviations seems to be “the name of the game”. This is not a urologic journal – do not use abbreviations – for example RALP, NS,NVB even in abstract – hopeless. The secondary end-points include variables to be used for power analysis – is 50 sufficient for this – see further below. And it would probably be better and more efficient if the full scale study was started and an interim analysis made for sample size (see further below). P 10 line 10 all patients with....CAN be approached. I think that should be “SHOULD” Page 10 line 44 : patients for should be “patients of Page 11 line 8: if surgeons have a minimum requirement of earlier procedures – why not radiologists?? Or do you plan for centralized radiology? Page 11 line 35 and following: the surgeon should be asked to state BEFORE operation the intention regarding nerve sparing I recommend that you also include in clinical record form filled out by the surgeon immediately after surgery the extent of lymph node dissection, For standard RALP the surgeon should answer questions after the operation not only about nerve sparing but also detailed questions about vesico-urethral anastomosis – as the technique can vary and this could impact on incontinence postoperatively Page 12 line 14+15: does the description “...the removal of the prostate for specimen painting, arrival of specimen in laboratory,...” refer to the frozen section – if not I suggest that these details are also given after standard RALP.
---

Basically all data collected about the atndard and interventional procedures should be the same – as it is my experience that you quite often wish to compare the two techniques, sometimes to udnerstnad better why differences occur

Page 12 line 35 – is “carius” correct or is it supposed to be “various”

Page 12 “Data collection” – it must be made clear the times after index surgery for these visits – seems to me to cover several years, as in most standard of care maps, a visit to give results of pathology and determine postoperative PSA is followed by a 12 months follow up? Very confusing without a point in time (related to index surgery).

To collect patient report by the care giver will ensure a “better” estimate of incontinence and erectile dysfunction than if the patients send them to a secretariat not immediately recognized as the “care giver”. Strongly advice against this.

And to ask the same questions as often as planned here, includes a risk of affecting the reports by “recall bias” – I suggest you reconsider as the secondary endpoint “incontinence” and “erectile dysfunction” may change slightly over time but it must surely be the “end-result” 12 or even 24 months postoperatively which is the truly important one, and not if a restitution appears 3 months earlier with the intervention. The plan is to ask about incontinence and erectile dysfunction and quality of life 5 times over 12 months – far to often I think for reliable answers. I would expect that both the RAND as well as the EQ5D instruments have recommendations about this, but regardless this is a well-known problem with “repeated measures” of this type. And on top of this the incontinence rate you (and I expect) 15-20% at 12 months based on two groups of 25 patients each, will mean not very many observations (4-5 per group). Can you really truxt your sample size calculation on this? I would hesitate, as the trial as such is a serios and expensive undertaking.

Health economic diary: my experience is that HE diaries are accurate but that the resource consumption reported there has little impact on total resource consumption, but adds a lot of work (to little or even no effect). The extra cost of the intervention tested (time in OR + extra “emergency” pathology) is such that the difference in patient reported costs has a very small chance of changing this. The extra resource use by the experimental intervention can be accepted but probably only by changing (substantially) the percentage of patients with permanent incontinence and/or erectile dysfunction and thus changing quality of life (QUALYS)

Page 13 line 55-56: Clavien-Dindo grading can vary depending on the grader, thus I strongly recommend one of the following two procedures: one and the same “grader” travels to the three sites and grades all randomized patients, preferably without access to randomization or that you set up for a quality control by re-grading performed by an outside expert on CD grading and comparison to the “trial internal grading”.

Page 13 line 59 and following into page 14: Pathology data – I recommend that you also include in clinical record form filled out by the surgeon immediately after surgery the extent of lymph node dissection that you state if you are going to collect number of lymph nodes, number of engaged lymph nodes etc. Further I think you should also have specified the competence of the individual pathologist, and have a standard document describing what the pathology report should include. One relatively common (described in the literature) problem is Gleason, where

	pathologists differ and thus the specificity/sensitivity is less than you would expect. Or do you plan for centralised pathology (even better if possible to use a highly competent pathologist), Page 14 – recurrence: I would expect very low percentage of recurrent (and at three months many would call it residual disease) disease, certainly less than 10% and thus your number of observations will be very few, and obviously not enough to use for any sample size calculation based on such an expected percentage out of 50 patients, divided into two groups. Regarding safety of data, ethics and so on, I do not comment as details like these have to conform to national laws and regulations. For this trial I presume that the EU normative regulations often called GDPR are followed as Britain still is a member of the EU. I cannot find any comment on the “fate of the feasibility part of the RCT – are you planning to include these 50 patients into the “full scale” trial, if you find that such a trial would be feasible? That is possible and could be regarded as “efficient”, but should be made clear in the protocol for the feasibility part.
--	---

REVIEWER	Antonio Galfano Niguarda Hospital, Milan, Italy
REVIEW RETURNED	11-Dec-2018

GENERAL COMMENTS	The idea and the need to perform a preliminary feasibility study before starting with the RCT is clearly understandable. The Authors should only add an explanation of the methods used to decide that 50 patients in 15 months were sufficient.
--

REVIEWER	Paolo Capogrosso IRCCS Ospedale San Raffaele Milan, Italy
REVIEW RETURNED	14-Dec-2018

GENERAL COMMENTS	The authors clearly reported a feasibility randomized study testing the effect of a novel frozen section technique in decreasing the risk of positive surgical margins and increasing the chance of a proper NS at RALP. The proposed surgical technique is interesting and I would be curious to see the results of this study. The protocol has been well described and does not seem to have major flaws.
---

REVIEWER	Philipp Dahm Minneapolis VAMC, USA
REVIEW RETURNED	03-Feb-2019

GENERAL COMMENTS	Ongoing efforts for a trial on nerve-sparing RALP are very laudable and this trial appears well designed, I have the following comments: Abstract: I would suggest you reword "RALP offers potential cure..."; we know that approximately 1 in 3 men with recur. Your statement appears overly optimistic.
--

	Strength and limitations: It is unclear to me what the word "robust" refers to. Would also reword "feasibility study not powered for definitive comparison of outcomes". Introduction: There is no mention of the use of MRI for preoperative planning although it does feature appropriately in the methods as a mandatory element of preoperative planning. I'd be sure to address here too. Trial design: I stumbled over the wording: "...researchers...will not routinely be informed...". I would make a stronger statement to indicate blinding of personnel other than surgeon/OR team. Exclusion criteria: I'd welcome additional detail as to how you will operationalize "unable to undergo RALP" or "nerve-sparing deemed futile...". This is rather vague. Maybe provide some examples as to the most common reasons for exclusion. Sample size: It would seem that the authors should have enough data from their own experience and the literature to have sense as to how many patients they might have to recruit for a definitive trial. To assess whether this pilot trial has any chance of success it would also be most helpful to have a sense as to how many RALP are performed at each of the three centers (nerve-sparing and non-nerve-sparing). Recruitment: This may be semantics but in the world I practice patients chose their treatment approach (such as RALP) if deemed appropriate candidate; I'd reword. Randomization: Please be even more explicit in describing that allocation will be concealed. Surgeon and unit accreditation: Having performed lots of case (here: 100 or more) is reassuring as an indicator of quality but appears insufficient. Maybe expand on expectations with regard to expected outcomes as recorded in the BAUS database. Endpoints: Please comment on the choice of the RAND 36; wouldn't EPIC be a superior tool? I would make it more explicit in the methods if and which outcome assessors will be blinded. For example, blinding the pathologists may not be feasible or require additional planning. Similarly, those rating the severity of complications should be blinded.
--	--

VERSION 1 – AUTHOR RESPONSE

Reviewer(s)' Comments to Author:

Reviewer: 1

Reviewer Name: Eva Haglind

Institution and Country: Department of Surgery, Sahlgrenska University Hospital, Göteborg, Sweden

Please state any competing interests or state 'None declared': None declared

Please leave your comments for the authors below

I think the title contains a language problem - feasibility study RCT – where “study” should be deleted, as trial is hidden behind RCT. A good example of “danger” with abbreviations.

Thank you. Title revised. Now without abbreviations. to: NeuroSAFE Robot Assisted Laparoscopic Prostatectomy versus standard Robot Assisted Laparoscopic Prostatectomy for men with localized prostate cancer (NeuroSAFE PROOF): protocol for a randomised controlled trial feasibility study.

Abbreviations seems to be “the name of the game”. This is not a urologic journal – do not use abbreviations – for example RALP, NS,NVB even in abstract – hopeless.

Abbreviations removed from title. Abbreviation in abstracts remain.

The secondary end-points include variables to be used for power analysis – is 50 sufficient for this – see further below. And it would probably be better and more efficient if the full scale study was started and an interim analysis made for sample size (see further below).

Details of protocol for full study, including the addition of interim analyses, to follow in the NeuroSAFE PROOF full study protocol and protocol manuscript.

P 10 line 10 all patients with....CAN be approached. I think that should be “SHOULD”

Thank you, sentence has been amended to: ‘

All patients who have a diagnosis of prostate cancer in whom the specialist MDT has recommended RALP as a treatment option will be eligible to be approached.’ Given numbers of trial team members, it is not possible to ensure all of the men seen in the respective centres MDTs will be approached. Detailed screening log and CONSORT information will be kept in line with rigorous but pragmatic trial governance and conduct.

Page 10 line 44 : patients for should be “patients of

Page 11 line 8: if surgeons have a minimum requirement of earlier procedures – why not radiologists?? Or do you plan for centralized radiology?

Thank you. Please see next paragraph. Addition of 'at least 2 years experience in prostate MRI' for additional clarity.

Page 11 line 35 and following: the surgeon should be asked to state BEFORE operation the intention regarding nerve sparing

I recommend that you also include in clinical record form filled out by the surgeon immediately after surgery the extent of lymph node dissection,

For standard RALP the surgeon should answer questions after the operation not only about nerve sparing but also detailed questions about vesico-urethral anastomosis – as the technique can vary and this could impact on incontinence postoperatively

The operative parameters captured in the trial database represent a clinical record form filled in by the surgeon immediately after surgery. Regarding lymph node dissection and vesico urethral anastomosis: agree that this is an important influencing factor. We have not collected this information during the feasibility study, however we will be amending our operation data input in the main trial to include these factors in addition to estimated blood loss, Rocco stitch performance, anterior reconstruction. Thank you.

Page 12 line 14+15: does the description "...the removal of the prostate for specimen painting, arrival of specimen in laboratory,..." refer to the frozen section – if not I suggest that these details are also given after standard RALP.

Basically all data collected about the standard and interventional procedures should be the same – as it is my experience that you quite often wish to compare the two techniques, sometimes to understand better why differences occur

Agreed, we are aiming to capture all the same data. However, the prostatectomy specimen in the standard arm does not go for immediate analysis in the histology lab, therefore there will be necessarily differences in data input. Final histological (permanent prostatectomy paraffin embedded) analysis and data captured is identical. Times are also collected for standard arm, see sentence above: 'Detailed times of starting the RALP and finishing the RALP are recorded on the day of surgery in order to calculate the length of each case.'

Page 12 line 35 – is "curious" correct or is it supposed to be "various"

Thank you.

Page 12 "Data collection" – it must be made clear the times after index surgery for these visits – seems to me to cover several years, as in most standard of care maps, a visit to give results of pathology and determine postoperative PSA is followed by a 12 months follow up? Very confusing without a point in time (related to index surgery).

Improved clarity of this section. Thank you.

To collect patient report by the care giver will ensure a "better" estimate of incontinence and erectile dysfunction than if the patients send them to a secretariat not immediately recognized as the "care giver". Strongly advice against this.

Patient reported outcomes collected in person by trial research associate, who is one of the clinical team.

And to ask the same questions as often as planned here, includes a risk of affecting the reports by "recall bias" – I suggest you reconsider as the secondary endpoint "incontinence" and "erectile dysfunction" may change slightly over time but it must surely be the "end-result" 12 or even 24 months postoperatively which is the truly important one, and not if a restitution appears 3 months earlier with the intervention. The plan is to ask about incontinence and erectile dysfunction and quality of life 5 times over 12 months – far to often I think for reliable answers. I would expect that both the RAND as well as the EQ5D instruments have recommendations about this, but regardless this is a well-known problem with "repeated measures" of this type. And on top of this the incontinence rate you (and I expect) 15-20% at 12 months based on two groups of 25 patients each, will mean not very many observations (4-5 per group). Can you really trust your sample size calculation on this? I would hesitate, as the trial as such is a serious and expensive undertaking.

Feasibility study, therefore we have not made power calculations for this preliminary phase. We are considering adding an additional analysis of functional outcomes at 24 months but this is not within the remit of this study.

Health economic diary: my experience is that HE diaries are accurate but that the resource consumption reported there has little impact on total resource consumption, but adds a lot of work (to little or even no effect). The extra cost of the intervention tested (time in OR + extra "emergency" pathology) is such that the difference in patient reported costs has a very small chance of changing this. The extra resource use by the experimental intervention can be accepted but probably only by changing (substantially) the percentage of patients with permanent incontinence and/or erectile dysfunction and thus changing quality of life (QUALYS)

Health economist is part of the NeuroSAFE trial team. Following completion of the Feasibility study we will amend/improve approach to Health Economics analysis and data captured.

Page 13 line 55-56: Clavien-Dindo grading can vary depending on the grader, thus I strongly recommend one of the following two procedures: one and the same “grader” travels to the three sites and grades all randomized patients, preferably without access to randomization or that you set up for a quality control by re-grading performed by an outside expert on CD grading and comparison to the “trial internal grading”.

Agreed. Central, blinded grading of CD complications would improve trial design. See inserted sentence: ‘The central trial management team will ask sites to submit complication data blinded by arm of treatment. This will be assigned Clavien-Dindo classification centrally.’ Page 9.

AGREE

Page 13 line 59 and following into page 14: Pathology data – I recommend that you also include in clinical record form filled out by the surgeon immediately after surgery the extent of lymph node dissection that you state if you are going to collect number of lymph nodes, number of engaged lymph nodes etc.

Lymph node dissection and whether positive for cancer or not is already included in operative and histological parameters, respectively. Numbers of nodes harvested not related to remit of this study.

Further I think you should also have specified the competence of the individual pathologist, and have a standard document describing what the pathology report should include. One relatively common (described in the literature) problem is Gleason, where pathologists differ and thus the specificity/sensitivity is less than you would expect. Or do you plan for centralised pathology (even better if possible to use a highly competent pathologist),

Grade of reporting histologist is already included. Have specified this more clearly: ‘Detailed outcomes of the FFS analysis is captured and included in the reporting, including number of sections positive, length of positive margin, identity and grade of pathologist.

Page 14 – recurrence: I would expect very low percentage of recurrent (and at three months many would call it residual disease) disease, certainly less than 10% and thus your number of observations will be very few, and obviously not enough to use for any sample size calculation based on such an expected percentage out of 50 patients, divided into two groups.

Feasibility study to rehearse the ability to perform this trial. Full trial protocol will take into consideration likelihood and rate of events for any oncological outcomes in the power calculations and size estimations. The preliminary data from this feasibility will be helpful in this endeavour, but we will not explicitly or specifically refer to this in this protocol.

Regarding safety of data, ethics and so on, I do not comment as details like these have to conform to national laws and regulations. For this trial I presume that the EU normative regulations often called GDPR are followed as Britain still is a member of the EU.

Correct.

I cannot find any comment on the “fate of the feasibility part of the RCT – are you planning to include these 50 patients into the “full scale” trial, if you find that such a trial would be feasible? That is possible and could be regarded as “efficient”, but should be made clear in the protocol for the feasibility part.

Thank you. Additional sentence ‘The decision to include the 50 feasibility study patients in the full-scale NeuroSAFE PROOF trial will only be allowed if the feasibility study aligns sufficiently closely, and will be at the discretion of the independent trial steering committee.’

Reviewer: 2

Reviewer Name: Antonio Galfano

Institution and Country: Niguarda Hospital, Milan, Italy

Please state any competing interests or state ‘None declared’: None declared

Please leave your comments for the authors below

The idea and the need to perform a preliminary feasibility study before starting with the RCT is clearly understandable.

Thank you.

The Authors should only add an explanation of the methods used to decide that 50 patients in 15 months were sufficient.

Thank you. Additional sentence inserted. Fifty men was decided upon in order to demonstrate that if similar recruitment rates were maintained in the full-scale NeuroSAFE PROOF study, the trial would be able to recruit the several hundreds of men likely necessary to appropriately power the study (pending a formal power calculation) over the course of approximately 3 years.

Reviewer: 3

Reviewer Name: Paolo Capogrosso

Institution and Country: IRCCS Ospedale San Raffaele - Milan, Italy

Please state any competing interests or state 'None declared': none

Please leave your comments for the authors below

The authors clearly reported a feasibility randomized study testing the effect of a novel frozen section technique in decreasing the risk of positive surgical margins and increasing the chance of a proper NS at RALP. The proposed surgical technique is interesting and I would be curious to see the results of this study. The protocol has been well described and does not seem to have major flaws.

Thank you. We look forward to presenting and publishing our results.

Reviewer: 4

Reviewer Name: Philipp Dahm

Institution and Country: Minneapolis VAMC, USA

Please state any competing interests or state 'None declared': None

Please leave your comments for the authors below

Ongoing efforts for a trial on nerve-sparing RALP are very laudable and this trial appears well designed, I have the following comments:

Abstract: I would suggest you reword "RALP offers potential cure..."; we know that approximately 1 in 3 men with recur. Your statement appears overly optimistic.

Agreed, thank you. Change.

Strength and limitations: It is unclear to me what the word "robust" refers to. Would also reword "feasibility study not powered for definitive comparison of outcomes".

Agreed, thank you. Change

Introduction: There is no mention of the use of MRI for preoperative planning although it does feature appropriately in the methods as a mandatory element of preoperative planning. I'd be sure to address here too.

Thank you. Additional sentence. 'In particular, pooled data from a recent diagnostic meta-analysis found MRI to have a limited sensitivity of 0.57(95% Confidence Interval 0.49-0.64) when predicting ECE.(6)' page 3

Trial design: I stumbled over the wording: "...researchers...will not routinely be informed...". I would make a stronger statement to indicate blinding of personnel other than surgeon/OR team.

Agreed. Rephrased to: 'Researchers for whom knowledge of allocation is imperative, ie those involved in operating on patients or coordinating operating lists or pathology teams are not blinded to treatment allocation, other members of the research team are blinded to treatment allocation.' Page 5 line 4.

Exclusion criteria: I'd welcome additional detail as to how you will operationalize "unable to undergo RALP" or "nerve-sparing deemed futile...". This is rather vague. Maybe provide some examples as to the most common reasons for exclusion.

Sample size: It would seem that the authors should have enough data from their own experience and the literature to have sense as to how many patients they might have to recruit for a definitive trial.

Correct. See comment from reviewer 2. This was part of the decision to have a feasibility target of 50 patients. We have included an additional sentence to give a sense of this without performing a formal power calculation/size estimation as this is beyond the scope of this paper: 'Fifty men was decided upon in order to demonstrate that if similar recruitment rates were maintained in the full-scale NeuroSAFE PROOF study, the trial would be able to recruit the several hundreds of men likely necessary to appropriately power the study (pending a formal power calculation) over the course of approximately 3 years.'

To assess whether this pilot trial has any chance of success it would also be most helpful to have a sense as to how many RALP are performed at each of the three centers (nerve-sparing and non-nerve-sparing).

(see below all centres perform over 250 cases per year and UCLH, lead site, performs over 700 per year)

Recruitment: This may be semantics but in the world I practice patients chose their treatment approach (such as RALP) if deemed appropriate candidate; I'd reword.

Randomization: Please be even more explicit in describing that allocation will be concealed.

Participants will not be informed of their treatment allocation until exiting the trial 12 months following their surgery. The clinical teams performing and coordinating surgery will not be blinded to treatment allocation as this is impractical, any members of the research team not involved in these activities will be blinded.

Surgeon and unit accreditation: Having performed lots of case (here: 100 or more) is reassuring as an indicator of quality but appears insufficient. Maybe expand on expectations with regard to expected outcomes as recorded in the BAUS database.

Additional clarity given 'Trial sites will have well-developed RALP programs; routinely performing at least 250 cases per year and undergoing satisfactory NHS quality assurance and safety visits.'

Endpoints: Please comment on the choice of the RAND 36; wouldn't EPIC be a superior tool?

We use Rand 36 and are unable to change this for the feasibility study at present.

I would make it more explicit in the methods if and which outcome assessors will be blinded. For example, blinding the pathologists may not be feasible or require additional planning. Similarly, those rating the severity of complications should be blinded.

Agreed. Similar comment from Reviewer1. Response:

Central, blinded grading of CD complications would improve trial design. See inserted sentence: The central trial management team will ask sites to submit complication data blinded by arm of treatment. This will be assigned Clavien-Dindo classification centrally. Page 9